

# Environmental modulation of the proteomic profiles from closely phylogenetically related populations of the red seaweed *Plocamium brasiliense*

Gabriela Calegario[1,2], Lucas Freitas[3,4], Eidy Santos[5], Bruno Silva[1,2], Louisi Oliveira[1,2], Gizele Garcia[1,2,6], Cláudia Omachi[1], Renato Pereira[7,8], Cristiane Thompson[1,2] and Fabiano Thompson[1,2]

[1] Institute of Biology, Federal University of Rio de Janeiro, Rio de Janeiro, Brazil
[2] SAGE-COPPE, Federal University of Rio de Janeiro, Rio de Janeiro, Brazil
[3] Department of Biochemistry, Federal University of Rio de Janeiro, Rio de Janeiro, Brazil
[4] Department of Genetics, Evolution, Microbiology and Immunology, State University of Campinas, Campinas, São Paulo, Brazil
[5] Unit of Biology, State University of the West Zone, Rio de Janeiro, Brazil
[6] Department of Undergraduate Education, Federal University of Rio de Janeiro, Macaé, Brazil
[7] Department of Marine Biology, Fluminense Federal University, Niterói, Brazil
[8] Rio de Janeiro Botanical Garden, Rio de Janeiro, Brazil

Corresponding author
Fabiano Thompson,
fabianothompson1@gmail.com

## ABSTRACT

The genus *Plocamium* encompasses seaweeds that are widely distributed throughout the world's oceans, with *Plocamium brasiliense* found along the tropical and subtropical coasts of the Western Atlantic. This wide distribution can lead to structured populations due to environmental differences (e.g., light levels or temperature), restricted gene flow, and the presence of cryptic species. Abiotic variation can also affect gene expression, which consequently leads to differences in the seaweeds protein profile. This study aimed to analyze the genetic and proteomic profiles of *P. brasiliense* sampled in two geographically distinct sites on the coastline of Rio de Janeiro state, Brazil: Arraial do Cabo (P1) and Búzios (P2). The genetic profiles of macroalgal specimens from these two sites were indistinguishable as assessed by the markers *UPA/23S*, *rbc*L, and *COI-5P*; however, the protein profiles varied significantly between populations from the two sites. At both sites the ribulose-1,5-biphosphate carboxylase/oxygenase was the most abundant protein found in *P. brasiliense* specimens. The number of phycobiliproteins differed between both sites with the highest numbers being found at P1, possibly due to water depth. The differences in proteomic profiles of the two nearly identical populations of *P. brasiliense* suggest that environmental parameters such as light availability and desiccation might induce distinct protein expression, probably as a result of the phenotypic plasticity within this population of seaweed.

## INTRODUCTION

Red seaweeds (Rhodophyta) are widely distributed around the world, inhabiting sites with different abiotic conditions, from polar to tropical regions (*Harley et al., 2012*; *Guiry & Guiry, 2016*). The genus *Plocamium* (Plocamiales, Rhodophyta) comprises 40 heterogeneous species found in temperate to tropical oceans. This seaweed genus is gathering attention due its importance in the production of bioactive natural products, for example, molecules with inhibitory properties against toxic effects of snake venoms (*Claudino et al., 2014*), anti-cancer (*Alves et al., 2018*), bone growth (*Carson & Clarke, 2018*), anti-herbivore, and herbicides (*Pereira & Costa-Lotufo, 2012*; *Pereira & Vasconcelos, 2014*). One example is the species *Plocamium brasiliense*, which occurs along the coasts of Brazil, Venezuela, Colombia, and the Caribbean Islands. This species is the only *Plocamium* found in Brazil and it ranges from the south of Espírito Santo state to Rio Grande do Sul, living in harsh environmental conditions such as desiccation stress and low temperatures (<10 °C). Morphology is highly variable among the 40 known species of the genus *Plocamium*, (*Cremades et al., 2011*).

The development of proteomics allows comparative studies of protein expression changes in response to changes in the environment (*Tee et al., 2015*; *Getachew et al., 2016*). Proteomic approaches are important tools to elucidate the mechanisms underlying the ability of seaweed species to tolerate a wide range of abiotic (*Contreras-Porcia & López-Cristoffanini, 2012*) and biotic conditions (*Getachew et al., 2014*). Proteomics is an important tool to determine the protein repertoire of marine holobionts, including seaweeds, sponges, and corals (*Garcia et al., 2016*). Previous studies have shown that the protein profiles of the brown seaweed *Saccharina japonica* are affected by seasonal changes (*Yotsukura et al., 2010*). Similarly, protein profiles of the kelp *Ecklonia cava* differ considerably at higher temperatures, suggesting temperature stress-induced changes (*Yotsukura et al., 2012*). Epizootic colonization of seaweed may also lead to changes in the biochemical composition of the seaweed host, which is reflected at the protein level (*Getachew et al., 2014*). However, protein extraction presents challenges because the biochemical and molecular characterization of seaweeds is limited compared to that of vascular plants and animals (*Contreras-Porcia & López-Cristoffanini, 2012*).

It is not clear how the genome of closely related populations of the same species respond to different environmental conditions generating distinct proteome profiles. The wide geographical distribution of *P. brasiliense* indicates a high phenotypic plasticity, which may be explained by variations at the transcriptomic (*De Oliveira et al., 2015*) and probably proteomic levels. In the present study we tested two hypotheses: populations of *P. brasiliense* that inhabit different geographic locations may be nearly identical genetically (first hypothesis), and the proteomes of these populations may differ in response to the environmental conditions (second hypothesis). We aimed to understand how variable abiotic conditions influence the proteome of *P. brasiliense* by analyzing protein expression profiles and the following molecular markers: universal plastid amplicon, domain V of the plastid 23S rRNA gene (*UPA/23S*), ribulose-1,5-biphosphate carboxylase/oxygenase (RuBisCO) large subunit (*rbc*L), and

cytochrome c oxidase subunit I (*COI-5P*). In addition, there were not previous studies on proteomics of this seaweed.

## MATERIAL AND METHODS

### Study area and sampling

Specimens of *P. brasiliense* were sampled at 10 m depth on August 22, 2013 in Cabo Frio Island (23°00′32″S, 42°00′27″W) in Arraial do Cabo (P1) and at two m depth on August 17, 2013 at Forno Beach (22°45′41″S, 41°52′32″W), located in Armação de Búzios (P2). The two sampling locations are approximately 30 km apart.

Cabo Frio Island is located within a marine protected area administered by Brazilian Navy. This site is considered the major site of upwelling on the coast of Rio de Janeiro, and the seawater temperature averages approximately 14 °C but can be as low as 11.8 °C (*Guenther et al., 2008*). Seaweeds at the Arraial do Cabo site were collected by SCUBA diving. The seaweed specimens were frozen in liquid nitrogen. Forno Beach is indirectly influenced by upwelling and is one of the most ecologically preserved regions in the state, with a rocky shore on both sides and a small (150 m) stretch of sand (*Yoneshigue, 1985*). The seawater temperature is normally approximately 18 °C (*Tâmega & Figueiredo, 2005*). *P. brasiliense* specimens were collected at the Búzios site by free diving. These sites were chosen due (1) the morphological differences between specimens from both sites, where Búzios had smaller individuals while Arraial had bigger individuals and (2) environmental differences due to upwelling, with Búzios experiencing less influence of this event while Arraial is located directly under the influence of upwelling. Seawater temperature was measured in situ, while chlorophyll *a* concentration was obtained by satellite data analyzes (MODIS aqua; A20121772013272). Chlorophyll *a* values were obtained every 8 days, from June 26, 2012 to September 27, 2013.

### DNA extraction, PCR, and sequencing

We collected ten specimens per site and the total DNA was extracted from the algal tissue using DNeasy® Plant Mini Kit (QIAGEN, Germantown, MD, USA). We purified the DNA with PowerClean® DNA Cleanup Kit (MO BIO, Carlsbad, CA, USA). PCR was performed using the Veriti® thermal cycler (Applied Biosystems, Foster City, CA, USA) and the primers UPA/23S (*Sherwood & Presting, 2007*), *rbc*L (*Lin, Fredericq & Hommersand, 2001*), and *COI-5P* (*Saunders & Lehmkuhl, 2005*) using the amplification conditions described by these authors. These primers were chosen because they showed good efficiency in distinguishing algae populations when used in previously population genetics of marine algae (*Lin, Fredericq & Hommersand, 2001*; *Saunders & Lehmkuhl, 2005*; *Sherwood & Presting, 2007*). *Sherwood et al. (2010)* showed that UPA/23S and COI-5P are useful markers for algal biodiversity surveys, while other studies demonstrate that *rbc*L is also a reliable molecular marker (*Yang et al., 2008*; *Tan et al., 2018*).

PCR products were purified using USB® ExoSAP-IT® PCR Product Cleanup (Affymetrix, Santa Clara, CA, USA) and then sequenced using an ABI 3500 automated sequencer (Applied Biosystems, Foster City, CA, USA), the same primers used for PCR reactions, and the ABI PRISM BigDye terminator v3.1 Cycle Sequencing Kit

(Applied Biosystems, Foster City, CA, USA). The sequences were assembled using CodonCode Aligner (CodonCode Corp., Centerville, MA, USA).

## Phylogenetic analysis

Publicly available sequences for UPA/23S, *rbc*L, and *COI-5P* from *Plocamium* spp., *Laurencia dendroidea* and *L. majuscule* (closest species with sequence available) were downloaded from GenBank (*Benson et al., 2017*). The total number of sequences used was 234, 36 of which were obtained in this study (Table S1). Sequences for each gene were aligned separately using the MUSCLE algorithm (*Edgar, 2004*) on SeaView4 (*Gouy, Guindon & Gascuel, 2010*), and edited manually with MEGA7 software (*Kumar, Stecher & Tamura, 2016*).

To determine the potential species to be tested, we first reconstructed the topology of the *COI-5P Plocamium* genus using the maximum likelihood (ML) approach in RAxML v. 8.1.17 (*Stamatakis, 2014*) with a GTR+$\Gamma$ substitution model and 1,000 bootstrap replicates. We then selected lineages with more than five specimens. Using this new dataset (143 sequences), we generated another ML tree using RAxML with the same parameters. For the multilocus dataset, we reconstructed the ML tree using RAxML and the same parameters. Further, we calculated the genetic differentiation between both populations using Nei's $G_{ST}$ (*Nei, 1973*).

## Protein extraction and separation

We selected six specimens of *P. brasiliense* collected in two closely related sites with distinct environmental conditions (P1 and P2, three specimens from each site) for protein extraction, SDS-PAGE, and identification. These procedures were carried out in two independent experiments (i.e., two replicates each) following a modified protocol (*Garcia et al., 2016*). The seaweed tissues were ground in liquid nitrogen using a mortar and pestle, and approximately 200 mg of the resulting powder was extracted using TRIzol® Reagent manufacturer's protocol (Life Technologies, Carlsbad, CA, USA). After the final ethanol wash, purified proteins were dissolved in resuspension buffer (7M urea, 2M thiourea, 4% CHAPs, 40 mM Tris). Quantification was performed using the 2D Quant kit (GE Healthcare, Little Chalfont, UK). We obtained protein concentrations of about two mg/mL from 200 mg of seaweed. After quantification, proteins were fractionated by SDS-PAGE using a Mini-PROTEAN® Tetra handcast system (Bio-Rad Laboratories, Hercules, CA, USA). Protein samples (40 µg) were mixed with resuspension buffer with 0.01% of bromophenolblue before loading onto gels. After electrophoresis, gels were stained with Coomassie brilliant blue R-250 (GE-Amersham Biosciences, Little Chalfont, UK). Whole lanes of the gels were sliced in bands, and proteins were digested with trypsin, as described previously (*Laemmli, 1970*; *Coelho et al., 2004*).

## Mass spectrometry and data analysis

The resulting peptides from trypsin digestion were extracted from gel slices and analyzed by LC-MS/MSin a Q-TOF quadrupole/orthogonal acceleration time-of-flight spectrometer (Micromass) (Waters, Milford, MA, USA) coupled to a Waters® nano-ultra performance liquid chromatography system (UPLC) (Waters Corp., Manchester, UK), as described in

*Garcia et al. (2016)*. Tryptic peptides were desalted and separated on Waters Opti-Pak C18 trap column and the C18 NanoEase 150 mm × 75 μm column, in the nanoACQUITY UPLC® 1D chromatography system (Waters, Milford, MA, USA). Instrument control and data acquisition were conducted by the MassLynx data system (Version 4.0, Waters) to perform the Data-dependent MS/MS acquisitions. The exact mass of each signal from the total ion current chromatogram was determined automatically using the Q-TofLockSpray™ (Waters, Milford, MA, USA).

All data were processed using ProteinLynx Global server (version 2.5, Waters), by using the lock spray reference ion to automatically correct the *m/z* scale on MS data. Data of each protein mixture obtained from gel slices were converted to peak list files (.pkl) by ProteinLynx software. Protein identification was performed using MS/MS Ion Search at Mascot Server (http://www.matrixscience.com/search_form_select.html; Matrix Science, London, UK) using NCBI-nr database. Mascot search parameters included one missed cleavage, cysteine carbamidomethylation, and methionine oxidation as fixed modifications, and a fragment mass tolerance of 0.3 Da. The raw Mascot server results were screened to recognition of the correct protein assignments. We considered real identifications, protein hits with Mowse scores higher than 62 (the threshold score for protein identification) with at least two distinct peptides (with at least 13 amino acid residues each) with reliable matches on protein sequence. Reliable matches (indicated as bold red matches by the Mascot Server) were characterized by a peptide score number higher than the threshold for each run, with an expect value (*P*-value) below 0.05 (<5% chance of being false).

Bioinformatics tools developed for protein identification are focused on single (model) organisms, and the identification of tryptic peptides is still laborious and time consuming. Studies focused on complex multispecies systems such as *Plocamium* holobionts demand the use of large sequence databases. Moreover, the often-limited amount of available data (genomes and transcriptomes) for this holobionts another important obstacle in metaproteomic studies and may lead to peptide misidentification.

To test for potential false identification, a decoy database analysis was included using the automatic decoy search in the Mascot Search Server (Matrix Science, London, UK), with at least two distinct peptide (>13 amino acid residues each) with reliable matches on protein sequence were one of the minimal thresholds to identify a protein. Reliable matches (indicated as bold red matches by the Mascot Server) were characterized by a peptide score number higher than the threshold for each run, with an expect value (*P*-value) below 0.05 (<5% chance of being false). A match is the highest scoring match to a particular query listed under the highest scoring protein containing that match. This means that protein hits with many peptide matches that are recognized as matches are the most likely correct assignments. Therefore, our proteome analyzes were properly represented in this study.

The automatic decoy database search performed by the Mascot server was used to determine the false discovery rate (FDR). Functional contribution to seaweed proteomes were assessed by grouping polypeptides that encompasses the same protein. Taxonomic categories of each polypeptide type were inferred according to the Universal Protein
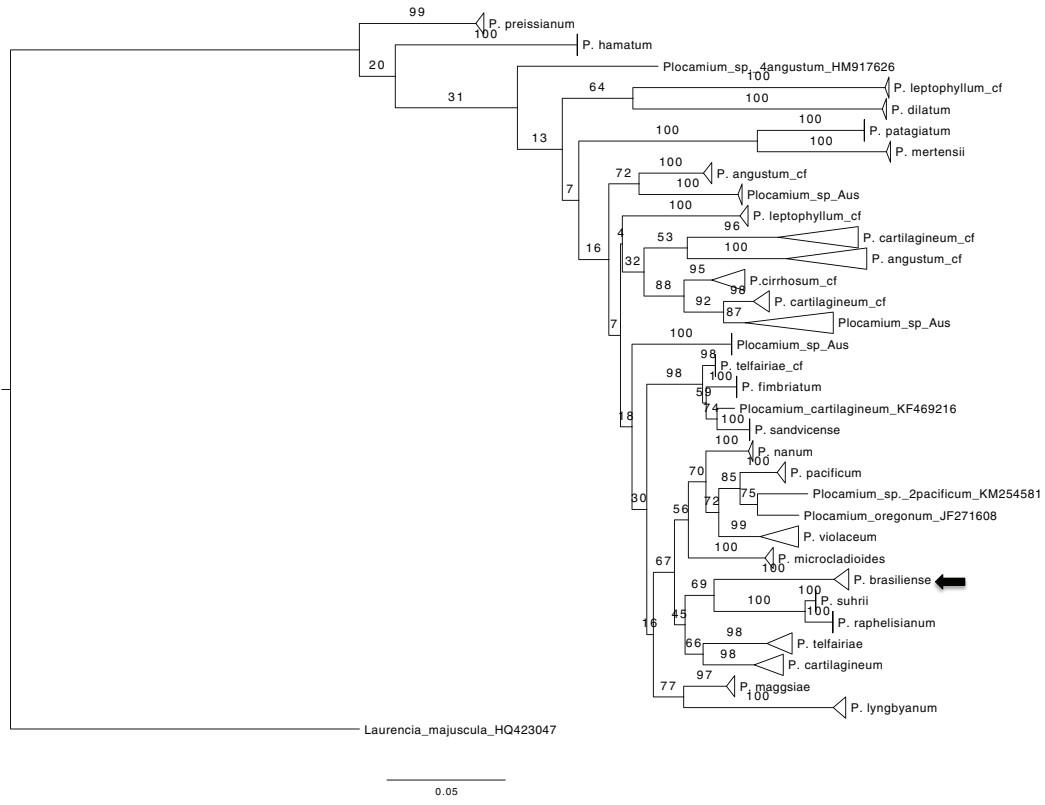

**Figure 1 ML tree from all COX1 *Plocamium* available at GenBank plus new sequences.** All groups >two individuals had a high bootstrap value. It is possible to note different groups composed by the "same species."

Knowledgebase (UniProt) (http://www.uniprot.org/uniprot/) classification. Taxonomic contributions were evaluated by clustering proteins from the same taxa. The relative contribution of each protein/taxon was calculated based on the percentage number related to the total identified in each condition. Then ANOVA test was applied to determine which proteins were significantly more abundant at the determined site ($P < 0.05$) using R v. 3.3.3 (*R Development Core Team, 2017*).

# RESULTS

The *Plocamium* populations studied here were obtained from two locations with different environmental conditions. Seawater temperature was lower in P1 (13 °C) than in P2 (23 °C), while chlorophyll *a* concentration was higher in P2 (mean: 1.15 mg/m$^3$; standard deviation: 1.91 mg/m$^3$; $N = 30$) than in P1 (mean: 0.81 mg/m$^3$; standard deviation: 1.80 mg/m$^3$; $N = 49$) ($P$-value = 0.0004653, $T$-test).

## *COI-5P* dataset

To obtain an overview of the relationships within *Plocamium* we used the *COI-5P* gene because of the large number of sequences available. The dataset consisted of 189 individuals and 656 bp. Our ML tree consisted of 29 groups (Fig. 1) containing at least two individuals

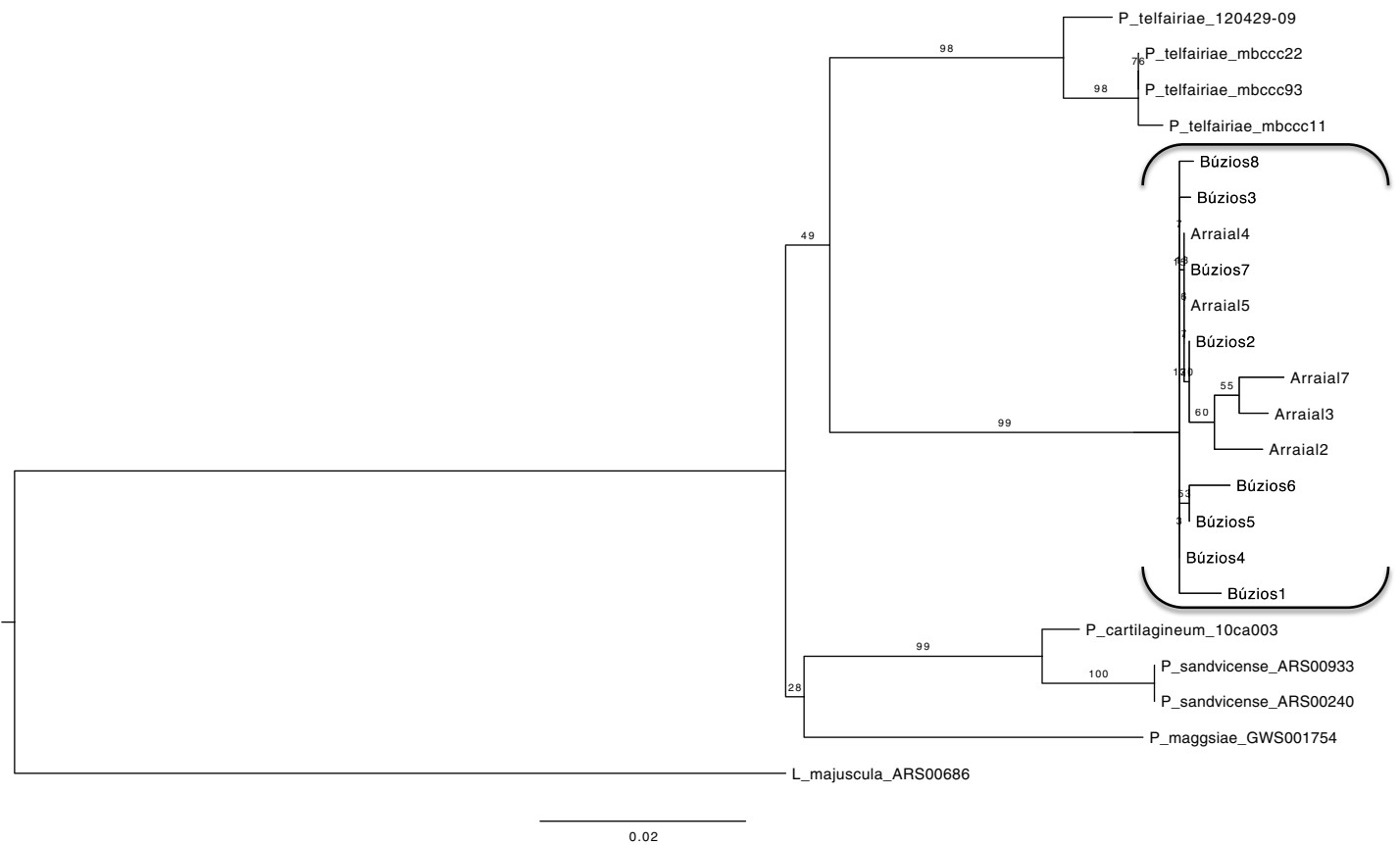

**Figure 2 Maximum likelihood phylogenetic tree for the concatenated dataset (individual level) showing the relationship of five *Plocamium* species.** In this tree, we did not observe a structure between the two populations of *P. brasiliense*.

with high bootstrap support values (>97; Plocamium_sp_Aus_2 was the only exception, with a bootstrap support value of 87). The relationships between groups and internal nodes of the tree did not have good bootstrap support values, and four individuals from different species (*P. angustum*, *P. cartilagineum*, *P. pacificum*, and *P. oregonum*) did not cluster with any group. The species sequenced in this work, *P. brasiliense*, which was sampled from two different sites, clustered into a single group, which was considered a sister lineage of a clade formed by *P. suhrii* and *P. raphelisianum*.

## Multilocus dataset

Using a dataset with three genes (*COI-5P*, UPA/23S, and *rbc*L) we reconstructed a concatenated ML tree with relationships between five species, where data of an individual was available for two or more genes (Fig. 2), allowing one gene to be missed in the final nucleotide matrix. This concatenated matrix consisted of 28 individuals and 1,073 bp. All species with more than one individual analyzed had their monophyletic status inferred with high bootstrap support value (>98). The sister lineage of *P. brasiliense* was inferred as *P. telfairiae*, and the low $G_{ST}$ (0.09) coupled with our concatenated analysis indicated lack of structure between populations at P1 and P2, rejecting the alternative

hypothesis (structured populations), thereby supporting the first hypothesis: individuals sampled in sites with different abiotic conditions do not form structured populations.

## Proteomic profile

The proteomes of six *P. brasiliense* specimens (three from Búzios and three from Arraial do Cabo) were extracted and visualized by SDS-PAGE. A total of 132 SDS-PAGE slices of protein bands were obtained (six biological specimens, two technical replicates for each specimen, Tables S2 and S3). Based on interpretation of Mascot searches against the NCBInr database, 2,791 polypeptides were identified from different taxa, encompassing 27 proteins. Average values of the FDR were about 1% varying from 0% to 7%.

Of the total specific proteins identified in holobionts *P. brasiliense* specimens collected from P1, 79.3% were annotated as belonging to Rhodophyta, followed by those belonging to Embryophyta (4.1%), Cyanobacteria (3.3%), Chordata and Mollusca (1.9% each), and Arthropoda and Chlorophyta (1.4% each). Within Rhodophyta, 35.5% of the identified proteins matched those of the genus *Plocamium* or the genera *Palmaria* and *Chondrus*, species phylogenetically close to *P. brasiliense*.

Of the total proteins identified in *P. brasiliense* specimens collected from P2, 82.2% were related to proteins annotated as belonging to Rhodophyta, followed by proteins belonging to Cyanobacteria and Embryophyta (6.2% each). Within Rhodophyta, 47.0% of the identified proteins matched those of the genus *Plocamium* or the genera *Chondrus* and *Gracilaria*, species phylogenetically close to *P. brasiliense*.

In the protein profiles of *P. brasiliense* specimens collected at P1 and P2, more than 20 bands were detected in each gel (each with three different biological replicates). From the 27 proteins five and four were significantly more abundant in P1 and P2, respectively (Table 1). Within the energy metabolism biomolecule category, the abundance of the RuBisCO was similar in both populations, while phycobiliproteins, especially phycocyanin, were significantly more abundant in the P1 population than in the P2 population (Table 1).

Among photosynthesis-related proteins, ATP synthase, which participate in ATP synthesis in the chloroplast, were detected in both populations of *P. brasiliense* but it was significantly more abundant in P1. Taken together, our findings indicate that *P. brasiliense* populations at P1 and P2 are non-structured populations; however, the behaviors of individuals are influenced by their abiotic environment, as indicated by the differential protein profiles in the two populations.

## DISCUSSION

### *P. brasiliense* populations from Búzios and Arraial have identical genetic profiles

Analysis of the phylogenetic markers (UPA/23S, *rbc*L, and *COI-5P*) in the two populations evaluated in this study demonstrate that they belong to the *P. brasiliense* species. The small $G_{ST}$ value (0.09) estimated between both populations in this study confirm our first hypothesis, that despite being collected from geographically distinct sites, the populations are indistinguishable. This finding is consistent with other studies evaluating

**Table 1 Proteins of *P. brasiliense* individuals sampled in two distinct locations at Rio de Janeiro state.**

| Protein | Arraial | Búzios | Total |
|---|---|---|---|
| 14-3-3 protein | 1 (0.12) | 2 (0.48) | 3 (0.24) |
| 40S ribosomal protein S4 | 3 (0.37) | 0 (0.00) | 3 (0.24) |
| Allophycocyanin* | 96 (11.76) | 68 (16.39) | 164 (13.32) |
| ATP synthase* | 48 (5.88) | 54 (13.01) | 102 (8.29) |
| ATPase | 13 (1.59) | 8 (1.93) | 21 (1.71) |
| CbbX* | 0 (0.00) | 4 (0.96) | 4 (0.32) |
| CfxQ | 3 (0.37) | 2 (0.48) | 5 (0.41) |
| Collagen | 3 (0.37) | 0 (0.00) | 3 (0.24) |
| Elongation factor* | 3 (0.37) | 8 (1.93) | 11 (0.89) |
| Fructose-1,6-bisphosphatase | 5 (0.61) | 2 (0.48) | 7 (0.57) |
| Glyceraldehyde-3-phosphate dehydrogenase | 9 (1.10) | 10 (2.41) | 19 (1.54) |
| Haloalkane dehalogenase* | 0 (0.00) | 6 (1.45) | 6 (0.49) |
| Heat shock protein 70* | 13 (1.59) | 2 (0.48) | 15 (1.22) |
| Histone protein | 3 (0.37) | 0 (0.00) | 3 (0.24) |
| Hypothetical protein* | 14 (1.72) | 0 (0.00) | 14 (1.14) |
| Paramyosin | 6 (0.74) | 0 (0.00) | 6 (0.49) |
| Phycobilisome | 2 (0.25) | 2 (0.48) | 4 (0.32) |
| Phycocyanin* | 80 (9.80) | 8 (1.93) | 88 (7.15) |
| Phycoerythrin | 235 (28.80) | 104 (25.06) | 339 (27.54) |
| Plastid oxygen-evolving enhancer 1 | 4 (0.49) | 4 (0.96) | 8 (0.65) |
| Ribulose-1,5-bisphosphate carboxylase | 257 (31.50) | 126 (30.36) | 383 (31.11) |
| Sulfate adenylyltransferase | 3 (0.37) | 0 (0.00) | 3 (0.24) |
| Translation elongation factor | 1 (0.12) | 1 (0.24) | 2 (0.16) |
| Tropomyosin | 6 (0.74) | 0 (0.00) | 6 (0.49) |
| Ubiquitin/actin fusion protein | 6 (0.74) | 0 (0.00) | 6 (0.49) |
| Uncharacterized protein | 2 (0.25) | 4 (0.96) | 6 (0.49) |

Notes:
The values are given in total number and percentage (within parenthesis).
* Abundance significantly distinct between sites.

Rhodophyta species. For example, *Gigartina skottsbergii* populations collected at sites up to 45 km apart were shown to be genetically homogeneous (*Faugeron et al., 2004*). Similarly, only a small genetic distance was observed between closely related individuals of distinct populations of the red alga *Gelidium canariense* separated by 0.5–21 km, with a positive correlation between geographic and genetic distances (*Bouza et al., 2006*).

Seaweeds are considered poor dispersers, because spores generally survive for only few days (*Santelices, 1990*). Indirect estimates indicate a low gene flow for seaweed species (*Wright, Zuccarello & Steinberg, 2000*), and suggest that short dispersal distances are significant in species differentiation (*Tatarenkov et al., 2007*). However, despite the presumed poor dispersal of spores, the survival of *P. brasiliense* spores may be a factor in the unstructured populations of *P. brasiliense*, even at a distance of approximately 30 km. Corroborating the spore dispersion in long distances by red algae, recently *Hu et al. (2018)*

estimated a $G_{ST}$ smaller than 0.1 for *Gracilariopsis lemaneiformis* populations distant by more than 200 km.

## Environmental modulation of proteomic plasticity in *Plocamium*

In this study, we aimed to analyze the *P. brasiliense* proteomes to disclose its modulation in response to abiotic conditions. In addition to seaweed proteome, this study also focused on complex multispecies systems associated to *P. brasiliense*, that demand the use of large sequence databases. We used a combination of Trizol extraction and 1D electrophoresis for protein isolation and separation, which yields protein extracts of high quality and a less laborious and faster methodology for a general description of the seaweed proteomic pattern.

The seaweed *P. brasiliense* is an autotrophic alga, which means that it obtains energy through photosynthesis. The major photosynthesis-related protein found in *P. brasiliense* specimens was RuBisCO, which was the most abundant protein in both sites, while the second most abundant group were phycobiliproteins. Thus, accumulation of these pigments may help harvest light.

Ribulose-1,5-biphosphate carboxylase/oxygenase is a bifunctional enzyme, that is, involved in both carbon dioxide fixation and oxygenation (*Wong et al., 2006*), creating a metabolic pathway in which photosynthesis and photorespiration compete (*Raines, 2011*). Further, the charge distribution on the active site of the RuBisCO, mainly the large chain may facilitate carbon uptake, enhancing survival and growth under stress conditions. For example, energy demand by *Plocamium* maybe higher in P2 due to the lower depth that may include periods of desiccation or nutrient-enrichment of seawater, as indicated by the higher chlorophyll *a* contents in the water of P2 observed here. As desiccation stress demands increased ATP, higher levels of chlorophyll proteins are needed to increase $CO_2$ fixation by RuBisCO (*Tee et al., 2015*).

The presence of heat shock protein 70 in both populations of *Plocamium* also suggest possible environmental stress responses, or simply responses related with housekeeping roles in cells (e.g., protein folding, assembly, translocation, and degradation) (*Hartl, 1996*; *Gerloff-Elias et al., 2006*). Other stress-related proteins, like vanadium-dependent haloperoxidases, were not found in our study.

Altogether phycobiliproteins were more abundant at P1, where the most abundant phycobiliprotein was phycoerythrin (27.36%), a possible response of the seaweed to higher depth. In Rhodophyta and Cyanobacteria, phycobiliproteins (antenna complex) are always associated with the photosynthetic apparatus (*Lemasson, De Marsac & Cohen-Bazire, 1973*; *Sinha & Häder, 2003*). These proteins are the major photosynthetic accessory pigments that absorb and transfer sunlight to chlorophyll *a* (*Bryant, 1982*). The number of phycobiliproteins detected in this study is consistent with results of previous studies, which showed that light wavelength and water temperature, both of which increase photosynthetic efficiency, directly affect the composition of phycobiliproteins (*Govindjee & Braun, 1974*).

Phycobiliproteins are classified into three major groups according to their absorption spectrum: phycoerythrins (540–570 nm), phycocyanins (610–620 nm), and

allophycocyanins (650–655 nm) (*Kannaujiya & Sinha, 2015*). *Plocamium* species possess chlorophyll *a*, and the presence of phycobiliproteins is important to capture light of various wavelengths, allowing for more efficient photosynthesis (*Govindjee & Braun, 1974*). Furthermore, the composition of photosynthetic pigments in seaweeds varies according to environmental parameters (*Bryant, 1982*).

The first collection site of *P. brasiliense* (P1) is a subtidal environment, and specimens were collected at a depth of 10 m and water temperature of 13 °C; these conditions may account for the physiologic mechanisms that allow this seaweed to adapt to varying light conditions. Specimens of *P. brasiliense* collected from the second site (P2) also express phycobiliprotein complex proteins but at a lower level. Previous studies showed that these are upregulated in the environmental conditions of higher temperature and desiccation (*López-Cristoffanini et al., 2015*), which may explain the expression of allophycocyanin in *P. brasiliense* specimens from P2, which were exposed to extreme sunlight and desiccation. Regarding the other two phycobiliproteins, phycoerythrin and phycocyanin, we suggest that they are found in higher numbers in P1 because shorter light wavelengths reach deeper depths than longer light wavelengths. The majority of proteins expressed in P2 were previously reported as stress response proteins: CbbX and Elongation Factor on nitrogen stress-response in the red algae *Gracilaria gracilis* (*Naidoo, Rafudeen & Coyne, 2016*), ATP synthase on light stress-response in the seagrass *Zostera muelleri* (*Kumar et al., 2017*) and Haloalkanedehalogenase on pollution stress-response (*Buryska et al., 2018*). These proteins are likely to be involved in stress-response in P2 individuals since this population is in a region with higher temperatures compared with P1, and is routinely under desiccation stress.

Although the protein profile was obtained from 1D semi-quantitative method, we analyzed specimens exposed to distinct abiotic conditions (in situ). This way our study brings new information on the proteome of Rhodophyta class, which until this study were restricted to controlled experiments in laboratories (*Wong et al., 2006*; *López-Cristoffanini et al., 2015*; *Naidoo, Rafudeen & Coyne, 2016*; *Xu et al., 2016*). Further this is a first attempt to uncover the proteomic diversity of the genus *Plocamium*.

## CONCLUSION

Protein content of two populations of *P. brasiliense* that are nearly identical genetically is modulated by environmental conditions. The high phenotypic plasticity of this red seaweed allows it to occupy contrasting ecological niches, from sites of upwelling to intertidal zones. The differential proteomic profiles of these two populations are consistent with findings of previous studies, which reported differences in secondary metabolite profiles of distinct populations of *P. brasiliense*. This study is a first attempt to study the metaproteome of the *P. brasiliense* holobiont to determine how nearly identical populations cope with changing environmental conditions. In our study, light availability appeared to influence the production of phycobiliprotein (light-harvesting antennae).

### Funding

This work was supported by CNPq, CAPES and FAPERJ. The funders had no role in study design, data collection and analysis, decision to publish, or preparation of the manuscript.

### Grant Disclosure

The following grant information was disclosed by the authors:
CNPq, CAPES and FAPERJ.

### Competing Interests

Cristiane Thompson and Fabiano Thompson are Academic Editors for PeerJ.

### Author Contributions

- Gabriela Calegario conceived and designed the experiments, performed the experiments, prepared figures and/or tables, authored or reviewed drafts of the paper, approved the final draft.
- Lucas Freitas conceived and designed the experiments, analyzed the data, prepared figures and/or tables, authored or reviewed drafts of the paper, approved the final draft.
- Eidy Santos conceived and designed the experiments, performed the experiments, authored or reviewed drafts of the paper, approved the final draft.
- Bruno Silva performed the experiments, approved the final draft.
- Louisi Oliveira approved the final draft.
- Gizele Garcia analyzed the data, approved the final draft.
- Cláudia Omachi analyzed the data, approved the final draft.
- Renato Pereira conceived and designed the experiments, contributed reagents/materials/analysis tools, approved the final draft.
- Cristiane Thompson contributed reagents/materials/analysis tools, approved the final draft.
- Fabiano Thompson conceived and designed the experiments, contributed reagents/materials/analysis tools, authored or reviewed drafts of the paper, approved the final draft.

### Field Study Permissions

The following information was supplied relating to field study approvals (i.e., approving body and any reference numbers):

Field sampling was approved by ICMBIO to Fabiano Thompson (license to collect number: ICMBIO 11175-1).

### DNA Deposition

The following information was supplied regarding the deposition of DNA sequences:
KM974715.1: https://www.ncbi.nlm.nih.gov/nuccore/KM974715.
KM974716.1: https://www.ncbi.nlm.nih.gov/nuccore/KM974716.
KM974717.1: https://www.ncbi.nlm.nih.gov/nuccore/KM974717.
KM974718.1: https://www.ncbi.nlm.nih.gov/nuccore/KM974718.

## Data Availability

Raw data are available in Supplementary Files.

## Supplemental Information

Supplemental information for this article can be found online at http://dx.doi.org/10.7717/peerj.6469#supplemental-information.

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
