# Peer review of "Environmental modulation of the proteomic profiles from closely phylogenetically related populations of the red seaweed Plocamium brasiliense"

_PeerJ, doi:10.7717/peerj.6469_

## Round 0.1 · original submission · Major Revisions

Please address all the critical issues raised by both reviewers and revise manuscript accordingly.

Reviewer 1 ·

Basic reporting

Authors have focused on studying proteomic profiles of P. brasiliense sampled at two geographically distinct sites at coastline of Brazil. Focus of this study was to see if there is any divergence of the P. brasiliense population due to adaptive process. Authors have chosen two different sites for investigation and they have used both genetic and proteomic approach for this study. Overall the writing of this manuscript was improved from previous version in terms of use of professional language. Authors need to check for spaces and over use of words in some sections. Major issues with the manuscript is the formats of the tree diagrams and labeling of some of the figures. Is this due to format issue from word to pdf or they are simply misrepresented? It is hard to tell what proteins the authors are looking at in Figure 4 due to format issues. Figure 1 legend looks like incomplete at the end. Literature work was improved from previous versions of the manuscript. Proteomic information studies on P. brasiliense is very encouraging however the interpretation of results is not so clear and conclusive.

Authors mentioned in line 85 about the difference in secondary metabolites of P. brasiliense due to different geographical areas and In the next sentence they speculate whether proteomics of close populations vary due to environmental conditions. Are these studies were done based on the secondary metabolite context? This is a very loose concept. Moreover, Vasconcelos et al group only studied about monoterpenes but they have not done extensive characterization of several other secondary metabolites such as flavonoids, alkaloids, phenols etc.

Experimental design

Authors have chosen both genetic and proteomic approaches for this particular study. Sampling site and the method of collection were properly described. Maximum likelihood approach to determine the species that are tested using Cox1 was well explained. As mentioned above the figure representations are not formatted appropriately and need to be corrected. Internal transcribed spacer (ITS) has been widely used as marker for molecular phylogeny, especially due to its high variabilities between closely related species. Did the authors considered this approach to identify the populations between both sites?

Authors used proteomic approach to identify the spices in Sites P1 and P2 using Mascot searches against NCBInr databases. It is clear that most of the population belong to Rhodophyta based on Figure 3. Authors are encouraged to cut short the results section for this part as they can simply write that they observed higher percentages of Rhodophyta species at each site. There is no need to explain all the results with other species. Consider this as an minor comment. They can easily use this extra space to give more comprehensive results of the particular proteins identified based on P. brasiliense. Authors are encouraged to show the SDS-PAGE gel pictures for the proteomic data of P. Brasiliense specimens collected at P1 and P2 sites with markers for reference figures in supplemental data. Did the authors performed techniques such as 2D gel electrophoresis and Iso electric focusing for determination of proteins. This is will be one more verification step to confirm the findings.

Validity of the findings

Authors focused on proteomic profiles of phylogenetically related populations of the red seaweed Plocamium Brasiliense to verify if there is any divergence due to abiotic conditions. For this they have chosen two particular sites with varied climatic condition such as temperature, upwelling, depth, etc., Genetic profiles of the populations in the two sites indicate they are identical and belong to P. Brasiliense. This result is in accordance to their initial hypothesis.

To further investigate the population they have used proteomic approaches to see if there is any divergence or changes in response to abiotic conditions. The presence of increased levels Rubisco (photosynthesis related protein) is not any of significant find. Of course, these algae are autotrophic and they require photosynthesis apparatus for survival. Authors also showed the presence of photosynthetic proteins, phycobiliproteins.
In results section authors mentioned there is significantly more abundance of ATP synthase in P1 than P2 site but Table 1 shows there is not so much difference in the percentages (5,59 (48) in P1 and 13, 01 (54) in P2). Please clarify if there something that reviewers are missing any other supporting data. Authors also failed to explain why there is more phycocyanin in P1than P2 site (Figure 4). Authors mentioned in line 333 that phycobiliproteins are more abundant in P1, at the same time in line 353 and 354 they mention that higher temperatures conditions upregulate phycobiliproteins which suggests P2 as it has higher temperature. They are contradictory as one example they say P1 has more phycobiliproteins and in other they say its P2. Authors are encouraged to explain this. As per figure 2 there is a slight increase of phycobiliproteins in P1 rather than P2. Authors also mentioned in line 352 that P2 has lower protein levels and at the same time in line 354 they say higher temperatures upregulate phycobiliproteins. This is confusing aspect from this study.

If the whole goal of this study is to learn about the proteomic profiles of red seaweed from field samples, this manuscript is important for future research. However there is still a lack of clarity if there is any huge divergence between the genetically closely related species due to environmental modulations at proteomic level. Indeed there is some difference in some proteins especially phycocyanin, actin, heat sock protein etc., based on Table 1 but authors need to explain why they see these differences.

Additional comments

Manuscript reports on studying environmental effects on proteomes of phylogenetically related Plocamium brasiliense. Authors have improved the writing part of the manuscript and also they have used professional language. However work needs to be done on some spaces, over use of terms, representation and labelling (format) of figures. Discussion part need to be in-depth.

Authors have successfully used proteomic approach to provide information of key proteins of the populations from both sites that have varying conditions in terms of temperature, upwelling, depth etc., Major weakness of this study is lack of depth in interpretation the results. Some of the proteins does not have huge variation between two sites and the ones that have significant variation were not discussed in detail of why this kind of variation in expression. Temperature might be one of the factor but it looks like more work needs to be done to address other potential causes.
It will be very Interesting to increase the number of sites and sample size and do in depth analysis of these populations.

Reviewer 2 ·

Basic reporting

The article is written in English but it needs some corrections, to make some passages clearer. In which concerns the background presented in introduction section I make some recommendations that I think essential to understand the importance of the study. For example, although the authors aim to examine the genetic profiles of Plocamium specimens through genetic markers they do not present the characteristics of these markers in terms of rate of evolution. It might be clear to phycologists, but it is not obvious to everyone else. For example, Cytochrome oxidase has enough variations to distinguish many metazoans species, however, it is very conserved to corals and sponges. Besides, they do not explain why they did not examine ITS sequences, since the literature show ITS are one of the most variable markers even to Plocamium species (Yano et al, 2006), and then, should fit better to (genetically and geographically) close populations.
(Yano, T., Kamiya, M., Murakami, A., Sasaki, H., & Kawai, H. (2006). BIOCHEMICAL PHENOTYPES CORRESPONDING TO MOLECULAR PHYLOGENY OF THE RED ALGAE PLOCAMIUM (PLOCAMIALES, RHODOPHYTA): IMPLICATIONS OF INCONGRUENCE WITH THE CONVENTIONAL TAXONOMY1. Journal of Phycology, 42(1), 155–169. doi:10.1111/j.1529-8817.2006.00178.x)
The article structure partially conforms the format. I suggest some parts should be relocated to better fit the format. (It is detailed further).
Not all figures and tables seem relevant. The horizontal axis of figure 4 is illegible and the information can be easily placed in the text. Also, it is not clear that differences between bars are statistically significant or not. Several proteins listed in Table 1 are not mentioned or discussed in the text.

Experimental design

The article meets the aims and scope of the journal and present a well-defined question. However, I do not consider the methods applied the best to test the hypothesis. If the sequences of molecular markers were all identical it not means than all specimens genomes are identical because different parts of genome are under different evolutionary forces. The similarity between DNA markers only allows to state that there is no evidence that population are different species, or they are populations with any level of differentiation.
Why ITS marker was not included, and other phylogenetic methods was not examined? How did the authors select the model of nucleotide substitution to each marker and what test was done to verify if the concatenated markers would result in a reliable phylogenetic signal?
In many parts of the text the authors affirm that populations are not structured, however, the methods and the number of samples hamper such analyses. To verify the level of population structure, population genetics offers many options of clustering analyses, which have more appropriate statistics that better fit to such purpose than phylogenetic inference.

Validity of the findings

The study shows original results on proteomics of Plocamium specimens from Brazilian coast. Plocamium specimens living in shallow water show more clorophyll a than those living at 10m depth. It is related to RuBisCO enzyme that needs more energy in desiccation stress and for that uses more clorophyl to CO2 fixation. It seems to be the main finding of the article. However, no data was provided about desiccation, such: how long it lasts and how frequent it occurs.
The table 1 presents a list of protein identified in proteome profiles. Although some of them are significant different expressed nothing is discussed about them, for example: Haloalkane dehalogenase.
The conclusions on population structure are not appropriate stated since the methods and statistics applied do not allow such conclusions. Phylogenetic signal of variable markers can, indeed, suggest differentiation of populations, cryptic species and genetic isolation. However, such evolutionary phenomena must be tested with appropriate statistics and population genetics methods, for example, clustering or network analysis (Greenbaum et al. 2016 – doi: 10.1534/genetics.115.182626). The data only allow not discard the null-hypothesis of non-differentiation between populations.

Additional comments

Here I present more detailed comments:

Line 87: “the proteome of closely related populations of the same species respond to different environmental conditions” the authors mean: the genome respond to different environment conditions generating different proteome profiles?

Lines 101-103: It fits better on Introduction, particularly in the paragraph that explain the importance of Plocamium.

Lines 129-132: I suggest create a paragraph in the introduction to explain the importance of chosen molecular markers and there, to explain the variability levels of the markers with examples of species that were distinguished by them, if it varies in different classes or orders of related algae.
Line 147: “the maximum likelihood (ML) approach and RAxML” Do you mean in RAxML?

Line 148: How do you choose GTR+Γ substitution model? Which software were used to determine the substitution model? This inference should be determined by the data (sequences used to constructo the tree). There are software that estimate the best model to each data set.

Lines 150-151: If substitution rates vary too much from each other may not be suitable to concatenate the sequences. It is important, before concatenation, to screen sequences for homogeneity of base substitution and other congruence parameters (Rampersad et al. 2014 – doi: 10.1186/2193-1801-3-614).

Lines 196-197: The phrase is not clear.

Line 203-204: “We used a combination of tools to overcome this hindrance.” With this statement I expected to find a discussion about it, but there is not. I consider this statement vague and I suggest explain better.

Line 239: Here and elsewhere in the text I find “groups”. It is not clear what is it about. Groups is applied to which taxonomic level? It refers to a specific clade in the phylogeny? Please, make it clear.

Line 247-249: It is confused. First it is stated that were used three genes in concatenated tree, then it is stated that individuals with two genes were also used. Please, make it clear.

Line 253-254: The methods do not allow to infer if population is or not structured. (Greenbaum et al. 2016 – doi: 10.1534/genetics.115.182626.)

Line 261/275: What do you mean with 27 specific types? Classes of proteins? Protein identities? Please, make it clear.

Lines 282-285: The data do not allow to state that populations are not structured.

Lines 288-304: This subsection presents information that should be introduced in Introduction section to explain from where comes the hypothesis that populations of Plocamium could be genetically distant. And in the Discussion section go back to the idea with literature examples contrasting and supporting the findings.

Lines 315-316: “slightly more abundant in P2 than in P1” – If this different is not statistically significant remove it. If it is statistically significant I suggest change slightly more abundant by significantly more abundant.

Lines 317-327: The order of the information does not help the reader to deduce the conclusion by himself. I suggest rewrite it.

Line 370: Metaproteome is not a term applied in this study since the protein analysed were only those from the Plocamium origin. The meta prefix is applied when you study a community.

---

## Round 0.2 · Major Revisions

Please carefully address all the remaining issues raised by the reviewers and revise your manuscript accordingly.

Reviewer 1 ·

Basic reporting

Authors have focused on studying genetic and proteomic profiles of P. brasiliense sampled at two geographically distinct sites at coastline of Brazil and to test if there is any divergence of the P. brasiliense population due to adaptive process.
The writing of the manuscript has been improved considerably from previous versions but they still need to address issues such as spacing in the text. Sometimes the words are written together which is a major issue in the manuscript. Introduction part has been well trimmed and written well. Discussion part has been improved with some minor issues still required to be addressed. I have given some minor comments and suggestions for the author comments section below.

Experimental design

Authors have not done any major changes in this methods section from previous version. No new methods have been included. However authors still did not provide details on why they did not consider the use of ITS marker for their study. Reviewer’s encourage to use such approach in the future experiments to test their first hypothesis.

Validity of the findings

Authors focused on proteomic profiles of phylogenetically related populations of the red seaweed Plocamium Brasiliense to verify if there is any divergence due to abiotic conditions. For this they have chosen two particular sites with varied climatic condition such as temperature, upwelling, depth, etc., Genetic profiles of the populations in the two sites indicate they are identical and belong to P. Brasiliense. This result is in accordance to their initial hypothesis.

To further investigate the population they have used proteomic approaches to see if there is any divergence or changes in response to abiotic conditions.
Authors mentioned in line 354 that, “Specimens of P. brasiliense collected from the second site (P2) also express these proteins but at a lower level. This sentence has to be refrased. Authors either need to mention that they have found proteins at P1 and mention about the similarities in P2. They did not mention anything about finding proteins in P1. It is clear the some protein are more abundant at P1 then P2. Authors encouraged to mention this before talking about P2 site.

Authors mentioned in Line 361 about the presence of stress response proteins in P2. Can authors speculate the reason of these stress response proteins? Is it due the lower depth and higher temperatures and increased desiccation? Authors are encouraged to give future direction of this research. Marine microalgae has shown to have many defense mechanisms (chemical defence) against microbial species. Some of the upregulation of proteins can be part of defense mechanisms. Authors are encouraged to give any examples of such in P. brasiliense

Additional comments

Authors have done original work on analyzing the proteomic profiles of P. brasiliense. They have observed differences in sites (P1 and P2) from which they have collected the specimens and furture proteomic analysis. Authors have shown that difference in depth, temperature and desiccation might influence the proteomic profiles of the same species.

Authors need to check for spacing in the text. It might be due to formatting error. Please address.

Line 97-99: Authors can replace the sentence at line 67 as it is more suitable at the early introduction.
Below are some of the major and minor comments.
Figure 1 legend still looks incomplete at the end.
Table 1 legend still looks incomplete at the end.

Reviewer 2 ·

Basic reporting

The present work is a proteome analysis of Plocamium brasiliense, a seaweed occurring in part of Brazilian coast. The study shows upregulated expression of allophycocyanin in the environment with higher temperature and occurrence of desiccation in Plocamium brasiliense.
The article is written in English in conformity to professional standards. It includes sufficient background and literature references. The structure of the article is conforming the format of standard sections, except for the contents of lines 283-286 which are in Results section and I consider being part of Discussion/Conclusion. It is necessary to place space between words in several places throughout the text. Figure 3 seems irrelevant and the label of the table seems incomplete.

Experimental design

The article not successfully define its hypothesis and because of that it has a conceptual misinterpretation of the result (stating the null hypothesis, when it should be discarded or not). In fact, I recommend reading literature related to concept of null / alternative hypothesis and falsification of null (initial) hypothesis. (For example: Banerjee A, Chitnis UB, Jadhav SL, Bhawalkar JS, Chaudhury S. Hypothesis testing, type I and type II errors. Ind Psychiatry J. 2009;18(2):127-31.)
The collection location P2 do not correspond to coordinates. Geographical coordinates show a location called “Praia do Forno” at Arraial do Cabo, not at Buzios. It is distant about 8km from P1 (not 30km as appointed).
The phylogenetic reconstruction was made with GTR+Γ nucleotide substitution model, however it seems to be an excessively complex model to be used in COI phylogeny. The nucleotide substitution model must be estimated according to the data set, especially when genes with different rates of evolution are concatenated. Currently, there are many options of software capable to estimate the best fit model. I recommend reading the following literature: Arenas M. Trends in substitution models of molecular evolution. Front Genet. 2015; 6:319. Published 2015 Oct 26. doi:10.3389/fgene.2015.00319

Validity of the findings

The authors state the null hypothesis (lack of structuring of populations) when it can only be discarded or not. I suggest correcting that to meet the formal scientific speech. Also, it is important correcting the information of collection locations.
The authors affirm that Gst had low values suggesting lack of structuring of populations. However, Gst and other structuring indicators are relative values. It varies according to the molecular markers and to the organism examined. I recommend giving more emphasis to this part of the discussion, if possible, by giving more examples of studies that used the same markers and closely related organisms.
At one point in the discussion, the authors mention that the focus of the work includes a multispecies complex in addition to the seaweed proteome. However, no results were presented or discussed in relation to any species other than Plocamium brasiliense. Also, in the Conclusion paragraph it is stated that the study was a metaproteome. The meta prefix is generally used for studies that encompass a set of species, which does not occur in the present work.

---

## Round 0.3 · accepted · Accept

All the critical issues raised by the reviewers were adequately addressed and the manuscript was amended accordingly. The revised version is acceptable now.

#